Thermotogota diversity and distribution patterns revealed in Auka and JaichMaa ‘ja ‘ag hydrothermal vent fields in the Pescadero Basin, Gulf of California

http://orcid.org/0000-0002-5835-0455 Peña-Salinas Manet E. 1 2 manet.pena@uabc.edu.mx
http://orcid.org/0000-0002-2361-5935 Speth Daan R. 3 4
http://orcid.org/0000-0003-3322-7108 Utter Daniel R. 4
http://orcid.org/0000-0002-9561-355X Spelz Ronald M. 1
http://orcid.org/0000-0001-6040-729X Lim Sujung 4
Zierenberg Robert 5
Caress David W. 6
http://orcid.org/0000-0001-9594-559X Núñez Patricia G. 2
http://orcid.org/0000-0002-3279-9764 Vázquez Roberto 2
http://orcid.org/0000-0002-5374-6178 Orphan Victoria J. 4 vorphan@gps.caltech.edu
1 Facultad de Ciencias Marinas, Universidad Autónoma de Baja California , Ensenada, Baja California , Mexico
2 Laboratorio de Astrobiología, Instituto de Astronomía, Universidad Nacional Autónoma de México , Ensenada, Baja California , Mexico
3 Division of Microbial Ecology, Center for Microbiology and Environmental Systems Science, University of Vienna , Vienna , Austria
4 Division of Geological and Planetary Sciences, California Institute of Technology , Pasadena, California , United States
5 Department of Earth and Planetary Sciences, University of California, Davis , Davis, California , United States
6 Science Division, Monterey Bay Aquarium Research Institute , Moss Landing, California , United States
Kormas Konstantinos
Electronic publication date: 2024 Aug 19
Publication date: 2024
Volume: 12
Electronic Location ID: e17724
Received 2023 Dec 6; Accepted 2024 Jun 20
Copyright: © 2024 Peña-Salinas et al.
Copyright year: 2024
Copyright holder: Peña-Salinas et al.
License: This is an open access article distributed under the terms of the Creative Commons Attribution License, which permits unrestricted use, distribution, reproduction and adaptation in any medium and for any purpose provided that it is properly attributed. For attribution, the original author(s), title, publication source (PeerJ) and either DOI or URL of the article must be cited.
License URL: https://creativecommons.org/licenses/by/4.0/

Keywords: Deep-sea biosphere, Thermophiles, 16S rRNA, Microbial ecology, ASVs, Biogeography

Funding: National Council of Humanities, Science and Technology of Mexico (CONAHCYT) UABC’s 401/1/C/13/23 NSF Ocean Sciences Postdoctoral Research Fellowship 2126631 National Science Foundation Center for Dark Energy Biosphere Investigations (C-DEBI) NASA’s Interdisciplinary Consortia for Astrobiology Research (ICAR) 80NSSC23K1357 Manet E. Peña-Salinas was supported through a scholarship awarded by the National Council of Humanities, Science and Technology of Mexico (CONAHCYT). Funds for this study were derived from UABC’s internal Project No. 401/1/C/13/23 awarded to Ronald Spelz. Daniel Utter was supported by an NSF Ocean Sciences Postdoctoral Research Fellowship (award 2126631). Funding for the expedition, molecular and geochemical analyses was provided by grants from the National Science Foundation Center for Dark Energy Biosphere Investigations (C-DEBI) and NASA’s Interdisciplinary Consortia for Astrobiology Research (ICAR 80NSSC23K1357) both to Victoria J. Orphan. Victoria J. Orphan is a CIFAR research fellow in the Earth 4D program. There was no additional external funding received for this study. The funders had no role in study design, data collection and analysis, decision to publish, or preparation of the manuscript.

==============================
Discovering new deep hydrothermal vent systems is one of the biggest challenges in ocean exploration. They are a unique window to elucidate the physical, geochemical, and biological processes that occur on the seafloor and are involved in the evolution of life on Earth. In this study, we present a molecular analysis of the microbial composition within the newly discovered hydrothermal vent field, JaichMaa ‘ja ‘ag, situated in the Southern Pescadero Basin within the Gulf of California. During the cruise expedition FK181031 in 2018, 33 sediment cores were collected from various sites within the Pescadero vent fields and processed for 16S rRNA amplicon sequence variants (ASVs) and geochemical analysis. Correlative analysis of the chemical composition of hydrothermal pore fluids and microbial abundances identified several sediment-associated phyla, including Thermotogota, that appear to be enriched in sediment horizons impacted by hydrothermal fluid flow. Comparative analysis of Thermotogota with the previously explored Auka hydrothermal vent field situated 2 km away displayed broad similarity between the two locations, although at finer scales (e.g., ASV level), there were notable differences that point to core-to-core and site-level factors revealing distinct patterns of distribution and abundance within these two sediment-hosted hydrothermal vent fields. These patterns are intricately linked to the specific physical and geochemical conditions defining each vent, illuminating the complexity of this unique deep ocean chemosynthetic ecosystem.

Introduction

The discovery of hydrothermal vents in the deep ocean is fundamental to understanding our planet and its extreme ecosystems. As microbes dominate the deep-sea biosphere, exploring the environmental factors influencing microbial diversity is essential for gaining insights into their distribution in terms of biogeographic patterns.

Nevertheless, there is still much to unravel concerning the composition and structure of these microbial communities and the biogeochemical factors influencing their high diversity (Meier et al., 2017; Reysenbach et al., 2020; Sogin et al., 2006; Speth et al., 2022).

The Gulf of California

The Gulf of California (GoC) stands as a distinctive environment for ongoing research. The exploration of transform faults and spreading centers has continuously revealed new hydrothermal vent fields (Goffredi et al., 2017, 2021; Michel et al., 2018; Negrete-Aranda et al., 2021; Neumann et al., 2017, 2023; Paduan et al., 2018; Peña-Domínguez et al., 2022; Prol-Ledesma et al., 2013, Prol-Ledesma, Arango-Galván & Torres-Vera, 2016; Soule et al., 2018). Noteworthy among these discoveries are the newly identified vent sites at the Alarcon Rise and the adjacent Pescadero and Tamayo transform faults, and the South Pescadero Basin in the Pescadero Basin Complex (inset Fig. 1; Clague et al., 2018; Paduan et al., 2018). Since their discovery, the study of these new vent sites not only deepens our understanding of the intricate geological processes but also provides crucial insights into the dynamic interplay between tectonics and deep marine ecosystems in the GoC.

Figure 1 (A) Bathymetric map at 40-m resolution of the Southern Pescadero Basin indicating the location of the newly discovered hydrothermal vent fields, Auka and JaichMaa ‘ja ‘ag, marked by yellow and red dots, respectively. The Inset provides an overview of the study area within the regional tectonic context of the Gulf of California, a narrow continental-margin rift system formed between the Pacific and North American plates. The Gulf of California is characterized by a series of narrow pull-apart basins (thick red lines) interconnected by transform faults (thin black lines). Southern Pescadero Basin is the deepest and southernmost basin among a set of three closely spaced basins collectively referred to as the Pescadero Basin Complex (PBC). The PBC is flanked by other hydrothermal vent fields in the Guaymas Basin (GB), the Alarcon Rise (AR), and the East Pacific Rise (EPR). (B) AUV multibeam bathymetry with a lateral resolution of 1-m overlaid on shaded 40-m resolution ship-surveyed bathymetry. The black box delineates the JaichMaa ‘ja ‘ag hydrothermal vent field and its surrounding area.

Data source credit: Schmidt Ocean Institute. Maps created by authors using the mapping tools in GeoMapApp (www.geomapapp.org) / CC BY / CC BY (Ryan et al., 2009), and the Geospatial data illustration software CANVAS.

The South Pescadero Basin and the newly discovered hydrothermal vent sites

The South Pescadero Basin is the smallest of three tectonic basins that collectively form the Pescadero Basin Complex (Bischoff & Niemitz, 1980; Ramírez-Zerpa et al., 2022). The Auka hydrothermal vent field was discovered in 2015 at a depth of 3,670 m. This vent field stands as the deepest hydrothermal vent system known to date in the eastern Pacific (Fig. 1; Paduan et al., 2018). The sediment-hosted chimneys in Auka consist of calcite and anhydrite, with recorded maximum fluid temperatures nearing ~300 °C (Goffredi et al., 2017). Various studies have been conducted to gain a comprehensive understanding of its geological, geophysical, and biological context (Goffredi et al., 2017; Negrete-Aranda et al., 2021; Paduan et al., 2018; Ramírez-Zerpa et al., 2022; Salcedo, Soto & Paduan, 2019, 2021).

Despite these efforts, the microbial diversity of the microorganisms inhabiting the sediments surrounding the hydrothermal vents in Auka has been relatively understudied until recent years (Espinosa-Asuar et al., 2020; Speth et al., 2022; Wu et al., 2022). Phylogenetic studies have unveiled highly diverse microbial communities in Auka, exhibiting a variety of metabolisms enabling them to thrive in extreme conditions. Interestingly, approximately 20% of these species overlap with those found in the nearby Guaymas Basin (Speth et al., 2022).

Notably, a novel hydrothermal vent field was recently discovered in 2018 in the South Pescadero Basin. The new vent field, named JaichMaa ‘ja ‘ag, was confirmed and sampled using the Remote Operated Vehicle (ROV) Subastian during expedition FK181031 aboard the R/V Falkor in the Gulf of California (Fig. 1; Schmidt Ocean Institute, 2018). Situated only ~2 km southeast from Auka, JaichMaa ‘ja ‘ag shares similarities with Auka in terms of the size and structure of the hydrothermal mounds, the temperature and composition of the hydrothermal vent fluids, and the dominant faunal communities. Baker et al. (2016) previously estimated the average distance between hydrothermal discharges (plumes or vents) along spreading ridges, typically ranging from 3 to 20 km. In this context, JaichMaa ‘ja ‘ag offers a valuable opportunity to assess the dispersion and spatial patterns of microbial communities between two closely located hydrothermal vent fields within the Southern Pescadero Basin.

Expanding on prior research concerning thermal conditions in the South Pescadero Basin (e.g., Negrete-Aranda et al., 2021) and predicted optimal growth temperatures (OGT) of Auka microbes (Speth et al., 2022), we conducted a detailed investigation to elucidate the impact of temperature on the composition and distribution of microbial communities within the recently discovered JaichMaa ‘ja ‘ag hydrothermal field.

In this study, we present the first microbial ecology analysis of the JaichMaa ‘ja ‘ag vent field, including a comparative assessment with Auka in terms of microbial diversity and distribution patterns in the south Pescadero Basin, utilizing 16S rRNA analysis. Additionally, we explored thermophilic microbial groups related to hydrothermal diffuse fluids within sediments, focusing on the generally thermophilic phylum Thermotogota. These findings offer valuable insights into our understanding of hydrothermal vent systems and the influence of biogeochemical drivers, such as vent fluid availability, on the formation of sediment microbial communities.

Materials and Methods

High-resolution bathymetry data

Bathymetry data at 40- and 1-m resolution, respectively, were collected and processed during the cruise expedition FK181031 in November 2018 (Figs. 1 and 2) using the R/V Falkor’s 30 kHz hull-mounted Kongsberg EM302 multibeam sonar, and MBARI’s D. Allan B. Autonomous Underwater Vehicle (AUV) equipped with a Reson 400 kHz 7125 multibeam sonar, an Edgetech 110-kHz chirp sidescan (e.g., Caress et al., 2019, 2022). Raw and processed bathymetry data collected are archived and openly accessible at the Marine Geoscience Data System, MGDS (https://www.marine-geo.org/).

Figure 2 Overview of the sampling area and selected frame grabs of the JaichMaa ‘ja ‘ag vent field and adjacent vent sites.

(A) High-resolution (1-m) bathymetric map of the study area. The JaichMaa ‘ja ‘ag vent field (bottom) comprises distinct mounds and chimneys, labeled with names*, extending 400 m in a north-south direction. Other vent features north of JaichMaa ‘ja ‘ag, including Maija awi and Juwak Yuum, are shown. Sampling locations of push cores collected during expedition FK181031, analyzed in this study, are marked. (B) Top of a carbonate mound, informally referred to as Abuelita (‘Grandma’), covered in a white and gray bacterial mat with discrete clusters of Oasisia sp tubeworms, Clavularia anemones and P. orphanae scale worms around a localized spot of hydrothermal fluid discharge (top right). (C) ROV SuBastian’s manipulator arm holding an airtight water bottle used to sample the hydrothermal fluids ponded in the roof of Tay Ujaa (‘Big Cave’), the largest central mound in the vent field. Note the reflection of light from coarse calcite crystals lining the the cavern walls. (D) Push coring in sediment-covered bacterial mat surrounded by a broad vesicomyid clam bed located between Tay Ujaa and Weey ‘kual vent sites. (E) Weey ‘kual (‘Red Hill’) displaying a rich colony of Oasisia sp capping the mound’s summit. *Most vents’ names are derived from ancient indigenous Yuman tribes’ words, inhabitants of Baja California, Sonora (Mexico), and California and Arizona (United States). *The name of the reported hydrothermal vent field of JaichMaa ‘ja ‘ag mentioned in this article has been updated to reflect the accurate representation in the Kiliwa language. This revision supersedes the earlier reference in Negrete-Aranda et al. (2021). Photo and data source credits: Schmidt Ocean Institute. Maps created by authors using the mapping tools in GeoMapApp (www.geomapapp.org) / CC BY, and the Geospatial data illustration software CANVAS.

Sampling

Using the ROV SuBastian aboard R/V Falkor, a total of 10 30 cm long sediment push cores were collected across the JaichMaa ‘ja ‘ag vent field during dives S0195, S0197, and S0199 ( Data S1). The sediments analyzed were specifically taken from areas covered with microbial mats and exhibiting visible hydrothermal fluid flow (Fig. 2). Pore water geochemistry was obtained from the same core used for DNA extraction where possible except for 11 cores, which were only sampled for DNA, lacking geochemical data. Subsequent to core collection, temperature-depth profiles were measured whenever possible in proximity to the coring sites. Temperature measurements were recorded for each site at depths ranging from 15 to 30 centimeters below the seabed (cmbsf). Post-recovery, all sediment cores were extruded and sectioned into 1–3 cm horizons onboard the ship for pore water geochemical analysis following (Speth et al., 2022). Furthermore, 2 mL of sediment from each horizon were subsampled with a sterile cutoff 1 cc syringe and immediately frozen at −80 °C for subsequent DNA analysis.

Sample collection permits were granted by la Dirección General de Ordenamiento Pesquero y Acuícola, Comisión Nacional de Acuacultura y Pesca (CONAPESCA: Permiso de Pesca de Fomento No. PPFE/DGOPA-200/18) and la Dirección General de Geografía y Medio Ambiente, Instituto Nacional de Estadística y Geografía (INEGI: Autorización EG0122018), with the associated Diplomatic Note number 18-2083 (CTC/07345/18) from la Secretaría de Relaciones Exteriores—Agencia Mexicana de Cooperación Internacional para el Desarrollo / Dirección General de Cooperación Técnica y Científica.

Geochemical analysis

Pore water was extracted from sediments using a pneumatic sediment squeezer (KC Denmark A/S, Silkeborg, Denmark) under an argon gas headspace following Green-Saxena et al. (2014). Approximately 15 mL of pore water was extracted for 1 cm horizons, and ~50 mL for 3 cm horizons. The extracted pore water was filtered through a 0.22 μm pore size filter. A volume of 0.25 mL was preserved in a 0.5 M zinc acetate solution for sulfide analysis and stored at room temperature and 0.25 mL was stored at −20 °C for ion chromatography. Ion chromatography (IC) was performed at the Environmental Analysis Center, Caltech, using a Dionex ICS-2000 system (Dionex, Sunnyvale, CA, USA) equipped with parallel anion and cation columns. Samples, diluted at a ratio of 1:50 in 18 MΩ deionized water, were introduced via an autosampler and passed through an LC-Pak polisher (MilliporeSigma, Burlington, MA, USA) in series. Subsequently, the samples were loaded onto a 10 μL sample loop for both the anions and cations channels. The columns and detectors were maintained at 30 °C. For ion separation, a 2 mm Dionex IonPac AS19 analytical column protected by a 2 mm Dionex IonPac AG19 guard column (Thermo Fisher Scientific, Waltham, MA, USA) was employed. A potassium hydroxide eluent generator cartridge was employed to generate a hydroxide gradient, which was pumped at 0.25 mL min−1. The gradient remained constant at 10 mM for 5 min, increased linearly to 48.5 mM at 27 min, and then further increased to 50 mM at 40 min. Suppressed conductivity was detected using a Dionex AERS 500 suppressor operated in recycle mode with an applied current of 30 mA. For cation analysis, a 4 mm Dionex IonPac CS16 analytical column paired with a 4 mm Dionex IonPac CG16 guard column was utilized. An eluent generator cartridge, using methane-sulfonic acid, produced a gradient delivered at a flow rate of 0.36 mL min−1. The gradient was held constant at 10 mM for 5 min, then non-linearly increased to 20 mM at 20 min (Chromeleon curve 7, concave up), and further increased non-linearly to 40 mM at 40 min (Chromeleon curve 1, concave down). A Dionex CERS 2 mm suppressor with an applied current of 32 mA was used for suppressed conductivity detection. Chromatographic peaks integration was performed using Chromeleon 7.2 software with the Cobra algorithm. Concentrations were determined with correlation to known standards. The detection thresholds were approximately 10 μM for bromide and thiosulfate, 50 μM for ammonium, and 100 μM for calcium, potassium, and sulfate. Magnesium had a detection threshold of 400 μM. The IC data produced pore water concentrations for fluoride, acetate, formate, chloride, bromide, nitrate, sulfate, thiosulfate, lithium, sodium, ammonium, potassium, magnesium, and calcium (Data S1). We focused on the magnesium and ammonium findings as correlates of hydrothermal fluid concentration (Edmond & Damm, 1983).

Sulfide measurements, following the Cline reaction assay (Cline, 1969), included sample dilution, Cline reagent addition, and a 1-h reaction at room temperature in 96-well plates. Measurements, compared to a replicated 12-point standard sulfide curve, were conducted using a Tecan Sunrise 4.2 plate reader. For additional procedural details, please refer to Speth et al. (2022).

DNA extraction, 16S rRNA sequencing, and bioinformatics

For the isolation of microbial genomic DNA, a Qiagen Dneasy PowerSoil kit (Valencia, CA, USA) was employed to extract DNA from 80 subsamples taken from the 10 sediment cores, along with samples from microbial mats collected in the JaichMaa ‘ja ‘ag hydrothermal vent field. The DNA purification process followed the manufacturer’s protocol for 0.25 g of sediment, incorporating a series of six cleaning solutions; except that the lysis of cells was performed utilizing MP Biomedicals FastPrep-24 (Irvine, CA, USA). For the PCR amplification, the targeted region was the V4-V5 segment of the 16S rRNA gene, utilizing the 515f/926r primer set (Walters et al., 2016) modified with Illumina (San Diego, CA, USA) adapters on the 5′ end (515F 5′-TCGTCGGCAGCGTCAGATGTGTATAAGAGACAG-GTGYCAGCMGCCGCGGTAA-3′ and 926R 5′-GTCTCGTGGGCTCGGAGATGTGTATAAGAGACAG-CCGYCAATTYMTTTRAGTTT-3′). PCR reactions were performed in duplicate for every sample using Q5 Hot Start High-Fidelity 2x Master Mix (New England Biolabs, Ipswich, MA, USA). The reactions were conducted in a 15 μL volume with annealing at 54 °C for 28 cycles. In cases where no product was obtained, the number of cycles was increased as indicated in the sample metadata (Data S1). Subsequently, the PCR duplicates were pooled and barcoded using Illumina Nextera XT index 2 primers, which included unique 8-bp barcodes (P5 5′-AATGATACGGCGACCACCGAGATCTACAC-XXXXXXXX-TCGTCGGCAGCGTC-3′ and P7 5′-CAAGCAGAAGACGGCATACGAGAT-XXXXXXXX-GTCTCGTGGGCTCGG-3′). Barcoded primers were employed for amplification using the Q5 Hot Start PCR mixture. The total reaction volume was 25 μL with 2.5 μL of product. Annealing occurred at 66 °C and the process was cycled 10 times. Purification of the products was performed using the Millipore-Sigma (St. Louis, MO, USA) MultiScreen Plate MSNU03010 with a vacuum manifold. Quantification was carried out with the Thermo Fisher Scientific (Waltham, MA, USA) QuantIT PicoGreen dsDNA Assay Kit P11496 on the BioRad CFX96 Touch Real-Time PCR Detection System. Equimolar quantities of barcoded samples were combined and purified using the Qiagen PCR Purification Kit 28104 with the addition of 15–20% PhiX. Samples were sequenced on a MiSeq (Illumina San Diego, CA, USA) with a 2 × 300 bp chemistry protocol (Laragen Inc., Culver City, CA, USA). All raw data are available at NCBI BioProject PRJNA1045058, accession numbers SAMN38428873 through SAMN38428956.

Microbial community identifications from Auka samples were previously published by Speth et al. (2022) and publicly available at NCBI BioProject PRJNA713414; these were combined with the newly obtained JaichMaa ‘ja ‘ag data for a combined analysis.

Amplicon sequence variants (ASVs) were obtained with DADA2 v.1.22 following the developer’s recommendations (Callahan et al., 2016). A reproducible workflow with full details is available as Data S2. In brief, primers were trimmed with cutadapt (v3.4) to a mean depth of 26,315 reads per sample, discarding read pairs lacking amplicon primers. Reads were then trimmed to 240 and 200 bp for forward and reverse reads, respectively, and reads with more than two expected errors were excluded. The pool = ‘pseudo’ flag was used in the dada denoising step to increase sensitivity for potentially rare organisms. Forward and reverse sequences were merged before removing chimeras with default settings, resulting in a mean of 12,256 pairs retained per sample. Taxonomy was assigned to each ASV with the IDTAXA function (Murali, Bhargava & Wright, 2018) in the DECIPHER package in R using a custom database of SILVA 138 expanded with previously obtained but unpublished full-length 16S sequences from methane seep sites. Contaminant ASVs were identified and removed with the decontam package using the ‘prevalence’ method (Davis et al., 2018); a total of 36 ASVs were removed this way, resulting in a final count of 26,162 ASVs. ASVs were named in rank order by family, e.g., Thermotogaceae__2 is the second-most abundant Thermotogaceae ASV. The next-lowest taxon level was used if the ASV could not be confidently assigned a family-level designation (Datas S3 and S4).

Statistical analysis

To assess the degree of similarity between the newly introduced JaichMaa ‘ja ‘ag hydrothermal vent system and its microbial communities with Auka (Speth et al., 2022), a Non-Metric Multidimensional Scaling (NMDS) analysis was performed. This analysis was based on Bray-Curtis distances calculated with the ASV relative frequency data aggregated at the family level. Juwak Yuum, another site outside of the vent field with low activity in the S. Pescadero Basin (see Fig. 2), was included in the analysis to account for any potential outliers. Confidence ellipses were calculated with the ordiellipse function from the vegan package (Oksanen et al., 2022) a 95% confidence level.

Following Speth et al. (2022), we used magnesium concentration as a proxy for mixing of hydrothermal fluids in the pore water as increased fluid temperatures result in a depletion of magnesium (Edmond & Damm, 1983). The relative frequency of reads of different phyla was then correlated with magnesium concentrations in pore fluids from sediments. Phyla were only included with at least 0.5% abundance in at least 75 samples. For the ASV-level analysis of Thermotogota, abundant and prevalent ASVs were defined as ASVs that accounted for at least 20% of total Thermotogota abundance in at least one sample and were detectable (>1% of Thermotogota) in at least 12 samples.

The phylogenetic tree combining Thermotogota ASV sequences and Genome Taxonomy Database (GTDB, Parks et al., 2022) reference genomes was constructed by identifying 16S rRNA genes in GTDB reference genomes with the anvi-run-hmms program (Eren et al., 2020), exported, and trimmed to the V4–V5 region with cutadapt v3.4. A single sequence was chosen arbitrarily from Thermotogota reference genomes if more than one 16S rRNA sequence was detected. Sequences were then aligned with muscle v5.1 (Edgar, 2022). A maximum likelihood phylogeny was then constructed using IQ-TREE v2.1.2 with 1,000 bootstraps and the model GTR+F+I+G4 as selected by ModelFinder. The resultant phylogeny was midpoint rooted and displayed with the R package ggtree (Yu et al., 2017).

Results

JaichMaa ‘ja ‘ag: a new hydrothermal vent field

The recently discovered JaichMaa ‘ja ‘ag vent field (Figs. 1 and 2) was confirmed and sampled using the ROV Subastian during expedition FK181031 aboard the R/V Falkor in the Gulf of California (Schmidt Ocean Institute, 2018; Data S5). Situated at approximately 23.9402°N, 108.8556°W, roughly ~2 km south of Auka, and at a depth of ∼3,700 m, this new vent field unveiled intriguing features. AUV mapping revealed a series of mounds emerging from the sediment-covered basin, showcasing both active and inactive zones of hydrothermal vent fluids (Fig. 2). JaichMaa ‘ja ‘ag hydrothermal vent field covers an area of approximately 0.05 km2 (70–100 m wide by 400 m long). The nomenclature JaichMaa ‘ja ‘ag, roughly translating to ‘Liquid Metal’ from the dialect of the Kiliwa ethnic group that continues to live in northern Baja California, was chosen to reflect the interplay of light on the interface of hydrothermal fluid ponded in the roof of a cavern and the reflections of coarsely crystalline calcite covering the cavern walls. JaichMaa ‘ja ‘ag shares similarities with Auka in terms of the size and structure of the hydrothermal mounds, the temperature and composition of the hydrothermal vent fluids, and the dominant faunal communities. The new field comprises a series of mounds/hills, with hydrothermal edifices reaching heights up to ∼25 m.

The northernmost mounds include Tay Mpáan (Big Sister) to the southeast and the smaller Muutp Mpáan (Little Sister) to the northwest, both featuring inactive or diffuse venting areas covered with bacterial mats and various marine organisms (Fig. 2A). Moving south there is a small hill of carbonate called Abuelita (Fig. 2B) with an actively venting area displaying gray and white bacterial mats on the north flank, accompanied by Oasisia tubeworm colonies, Ostiactis anemones (Goffredi et al., 2021), Peinaleopolynoe orphanae and scale worms (Hatch et al., 2020), largely consistent with the fauna described previously from Auka (Goffredi et al., 2017). The largest central mound in the hydrothermal field is Tay Ujaa (Big Cave), a substantial mound with flanges where the hydrothermal fluid is ponded at the roof, forming a distinct optical interface (Fig. 2C). Between this mound and the next mound to the south, the area is populated by a vast field of clams buried in sediments and various bacterial mats. The southernmost feature is Weey ‘kual (Red Hill), a mound named for the exuberant coverage with Oasisia tubeworms around its summit (Fig. 2E).

Another noteworthy site in the vicinity of JaichMaa ‘ja ‘ag is Juwak Yuum (Two Eyes; Fig. 2A), located approximately 700 m to the N-NE. The site comprises two adjacent circular depressions, each measuring 3 m in depth and width, one of which was discharging shimmering water. The area is covered with sediment and adorned with patches of blue and yellow bacterial mats, as well as frenulates and anemones on the slopes, and Oasisia tubeworms at the bottom.

Furthermore, about 500 m east of Juwak Yuum, there is Maija awi, an isolated actively venting mound located midway between Auka and JaichMaa ‘ja ‘ag (Figs. 1B and 2A). Named after the divine serpent of water in the creation myth of the indigenous Kumiai culture, Maija awi was first explored during dive S0473 of the ROV Subastian on expedition FK210922 aboard the R/V Falkor in 2021 (Caress, Paduan & Institute, 2023; Schmidt Ocean Institute, 2018). This mound teems with life, dominated by polychaetes (Oasisia colonies, Dorvilleids, and Peinaleopolynoe orphanae scale worms), and Ostiactis anemones. The prominent chimney exhibits bright white hydrothermal calcite-anhydrite precipitates in the form of a dragon’s head, inspiring the vent’s name. Surrounding this unique structure are delicate spires and numerous flanges with shimmering water. Samples from this site are not included in the present article.

Overlapping microbial communities in the Southern Pescadero Basin vent fields

Microbial community composition was investigated at each site by sequencing the V4-V5 region of the 16S rRNA gene from sediments taken from JaichMaa ‘ja ‘ag and combined with previously published data from Auka (Speth et al., 2022). Each core was sectioned into 1–3 cm horizons from which DNA was extracted separately. Cores were generally dominated by microbes known to be associated with hydrothermal vents, such as Halobacterota, Desulfobacterota, Thermoplasmatota, Crenarchaeota, and more (Fig. S1). However, substantial compositional variation was observed across the dataset, leading us to investigate potential differences within and across the two vent fields.

We conducted a Non-Metric Multidimensional Scaling (NMDS) on family-level abundances. This ordination revealed a distinct cluster where most Auka sites (Diane’s vent, Matterhorn, north and south sides of Z vent), and JaichMaa ‘ja ‘ag sites (Abuelita and the chemosynthetic clam field between Tay Ujaa and Weey ‘kual) converged, indicating a high level of similarity between microbial community composition across the two vent fields (Fig. 3A). Within this analysis, the south side of Z vent in the Auka vent field shows the greatest scatter between data points, likely due to the larger number of samples from this site (n = 96 samples from 13 pushcores; Data S5). This variation is likely attributed to intra-core variation in community composition associated with core depth and steep physico-chemical gradients (as previously reported in Speth et al., 2022). The microbial communities from the low activity Juwak Yuum location plotted separately from samples collected within the hydrothermally active areas (Fig. 3A).

Figure 3 Non-metric multidimensional scaling (NMDS) plot based on 16S rRNA amplicon sequence variants of bacteria and archaea across the two vent sites Auka and JaichMaa ‘ja ‘ag, and the third site Juwak Yuum in the Pescadero Basin.

Ellipses denote 95% confidence limits for each location based on the observed samples. (A) Colors indicate the site from which the sample was collected during FK181031. (B) Colors indicate magnesium concentrations (mM).

As hydrothermal fluids tend to exhibit lower magnesium concentrations compared to seawater, here we use lower magnesium levels as a proxy for the increased proportion of hydrothermal fluid within sediment pore waters (Von Damm et al., 1985; Data S5). Magnesium concentrations were superimposed on the NMDS plot, showing a strong correlation between the variation in community composition and magnesium concentration (Fig. 3B).

Diffuse fluids revealed abundant thermophilic groups in the S. Pescadero vent fields

To identify the taxonomic group(s) associated with vent fluid, we examined the correlation between phylum-level relative frequency of reads and pore water magnesium concentration (Datas S4 and S5). The correlation coefficients were used as preliminary indicators to identify candidates for closer investigation. Top candidates included Desulfobacterota, Crenarchaota, and other putative thermophilic phyla (e.g., Thermoplasmatota, Halobacterota, Nanoarchaeota, and Caldatribacteriota) previously reported to be associated with active hydrothermal vents, including the neighboring sedimented vent system in Guaymas Basin (Fig. S2; Dombrowski, Teske & Baker, 2018; Huber, 2006; Laso-Pérez et al., 2023; Murphy et al., 2021; Shiotani et al., 2020; Speth et al., 2022; Teske, Callaghan & LaRowe, 2014; Zeng, Alain & Shao, 2021). Other community members were affiliated with microbial taxa described from other sedimented hydrothermal ecosystems, including sulfate-reducing bacteria, anaerobic methanotrophic archaea (ANME), and various taxa capable of sulfur oxidation, hydrogen oxidation, iron oxidation, sulfate reduction, and methane oxidation, among others (Fig. S1; Data S4).

In both Auka and JaichMaa ‘ja ‘ag, the Thermotogota phylum was highly ranked by correlation coefficient and occurred across a range of magnesium concentrations. At both vent sites, the Thermotogota relative frequency of reads was highest in samples with low magnesium, i.e., samples inferred to have higher levels of mixing with vent fluid. Interestingly, magnesium concentrations at Auka started at low concentrations near 1 mM, whereas at JaichMaa ‘ja ‘ag they started at 20 mM (Fig. 4). Such overlapping but distinct vent fluid regimes may influence the representation of different lineages of Thermotogota across the vent fields. As the ecology of the Thermotogota has not been extensively studied in hydrothermal vent sediments, we chose to highlight this phylum as an example of the dynamics within microbial communities in hydrothermal sediments across the different sites of the S. Pescadero Basin sites.

Figure 4 Relative frequency of reads for the phylum Thermotogota at Auka and JaichMaa ‘ja ‘ag.

Relative frequency of reads corresponding to Thermotogota plotted against magnesium concentration (mM) in pore water sediment samples from the (A) Auka, and (B) JaichMaa ’ja ’ag hydrothermal vent fields. Color coding denotes ammonium concentrations (mM) in the pore water.

Spatial patterns of Thermotogota sequence variants

The phylum Thermotogota was commonly recovered in sediments from both hydrothermal vent fields and shown to be strongly correlated with low magnesium concentrations (our hydrothermal fluid proxy). Here, we examined the diversity within this deeply branching bacterial lineage, along with their relative frequency of reads and distribution patterns across the Southern Pescadero vents. Of the total of 26,162 microbial ASVs, 153 unique ASVs were affiliated with Thermotogota across all sites. Specific tracking of all Thermotogota ASVs showed high variation both with core depth and site location for some ASVs, ranging from below detection to up to 32% of the total reads in a sample (mean of 2.93 ± 5.1% out of 135 samples). Members of the Thermotogaceae family were most diverse with 48 ASVs, followed by Mesoaciditogaceae (32 ASVs), Kosmotogaceae (16 ASVs), Fervidobacteriaceae (11 ASVs), and finally Petrotogaceae (4 ASVs). The remaining 42 Thermotogota ASVs could not be confidently assigned a family-level designation. To bolster the taxonomic assignment, we constructed a phylogeny using the Thermotogota ASV sequences and reference 16S rRNA sequences obtained from the Genome Taxonomy Database (GTDB, Parks et al., 2022) species references for Thermotogota (Fig. S4). ASV and GTDB sequences were generally consistent by taxonomy. The most notable exception is the apparent paraphyly of Thermotogaceae; however, this observation could be due to the limited information available in the short V4–V5 region as well as the fact that no GTDB references for Thermotogaceae contained suitable 16S rRNA genes. Nonetheless, the phylogenetic comparison of Auka and JaichMaa ‘ja ‘ag ASV suggests substantial phylogenetic diversity, underscoring the importance of exploring the evolution and ecology of these new hydrothermal vent sites.

These Thermotogota families were represented throughout the sampled regions of S. Pescadero Basin, however distinct distribution patterns were observed within the Auka and JaichMaa ‘ja ‘ag vent sites. 18 ASVs display the highest abundance and prevalence across the different cores obtained from the S. Pescadero Basin (Fig. 5A). These ASVs belong to the families Mesoaciditogaceae, Kosmotogaceae, Thermotogaceae, and Fervidobacteriaceae. In general, these 18 ASVs are extensively distributed across all sites within both hydrothermal vent fields, with their relative frequency of reads varying in relation to the sediment depth.

Figure 5 Microbial structure and composition of the phylum Thermotogota in the Auka and JaichMaa ‘ja ‘ag hydrothermal vent fields.

(A) Heatmap showing abundances of the 18 most abundant and prevalent Thermotogota amplicon sequence variants (rows) in each sediment horizon (columns), grouped by site, dive, and pushcore (nested columns). Note that the relative frequency is re-normalized to total Thermotogota abundance in a sample to better identify shifts among Thermotogota ASVs. (B) Heatmap showing the pairwise percent identity of the V4–V5 region of the 16S rRNA gene for the ASVs shown in (A). The dendrogram clusters the Bray-Curtis distances computed on the percent identity matrix (hclust method = ‘complete’). Abbreviations: PC, Pushcore; cmbsf, centimeters below seafloor.

As shown in Fig. 5A, the ASVs from Mesoaciditogaceae exhibit a significantly higher level of prevalence at the South of Z vent and Abuelita locations within a range of 0 to 17 cmbsf. When detected, Mesoaciditogaceae ASVs tended to dominate other Thermotogota, and occurred across all core depths, particularly in samples from the Auka vent field. Furthermore, Mesoaciditogaceae_1 and Mesoaciditogaceae_2 rarely co-exist in the same samples despite occurring in similar depths in different cores obtained from the same site. For instance, in the case of cores south of Z vent at Auka, where Mesoaciditogaceae_1 is abundant, Mesoaciditogaceae_2 is not, and vice versa. This down-core consistency persists despite drastically different magnesium concentrations, e.g., Mesoaciditogaceae_1 in PC6 from dive S0196 spanned nearly the full range of magnesium concentrations. Over this gradient, the abundance of Thermotogota relative to the entire community ranged from approximately 0.5% to 30% (Fig. S1), with one ASV (a Mesoaciditogaceae) dominant among the Thermotogota ASVs despite the presumed difference in hydrothermal fluid mixing.

In contrast, other Thermotogota ASVs peaked in abundance at the surface and diminished with depth, or vice versa (Fig. 5A). Most frequently, the ASV was replaced with an ASV from a different family, e.g., Kosmotogaceae_1 and Thermotogaceae_2 at Tay Ujaa and Weey ‘kual. Thermotogaceae ASVs tended to be more abundant at depth than at the surface, e.g., Thermotogaceae_1 in PC6 from Abuelita, Thermotogaceae_2 in PC1 and PC2 from Tay Ujaa and Weey ‘kual, etc. However, there were exceptions to this trend, such as in PC5 collected from a blue-tinged mat on the flank of Diane’s vent in Auka, where Thermotogaceae_1 was abundant in the first few sediment horizons and disappeared with depth.

Eight of Thermotogota ASVs were abundant and prevalent yet could not be assigned confidently to a taxon (Thermotogae_1 through Thermotogae_8). By using the percent identity of the sequenced 16S rRNA V4-V5 region as a rough proxy for their phylogenetic relatedness, we found that all were more similar to Thermotogaceae ASVs (sharing approximately 90% sequence identity) than to Kosmotogaceae or Mesoaciditogaceae ASVs (Fig. 5B). These ASVs also illustrated down-core dynamics, most notably in PC4 from north of Z vent where Thermotogae_4, Thermotogae_5, and Thermotogae_6 were all detectable in the same core but with abundance peaks at different depths (Figs. 5A and S3). Interestingly, Thermotogae_4 and Thermotogae_6 were closely related (>99% similarity over the sequenced region) but never were both abundant in the same horizon.

Discussion

In this study, we introduce the discovery of JaichMaa ‘ja ‘ag, the second hydrothermal vent field located within the S. Pescadero Basin in the Gulf of California. The relatively close proximity of JaichMaa ‘ja ‘ag and the previously described Auka site, separated by 2 km, enable comparison of the degree of overlap in microbial community structure. Recent 16S rRNA and metagenomic characterization of the microbial diversity within the Auka vent field revealed substantial overlap at the species level between the archaea and bacterial lineages and viruses from Auka sediments and those previously described from sedimented vent system from Guaymas Basin, approximately 400 km to the north of the S. Pescadero Basin site (Laso-Pérez et al., 2023; Speth et al., 2022).

The overall similarity of the bulk microbial community shared between Auka and JaichMaa ‘ja ‘ag fields is consistent with previous vent field comparisons (Anderson et al., 2017; Flores et al., 2011; Namirimu et al., 2022; Speth et al., 2022; Urich et al., 2014). Additionally, the intra-field variability in the sediment microbial assemblages observed at JaichMaa ‘ja ‘ag and their relationship with vent fluids (Mg concentration) suggest a strong response to the local physico-chemical conditions as previously noted by Flores et al. (2012). Heat-flow measurements support Auka and JaichMaa ‘ja ‘ag as part of a connected system that shares a deep hydrothermal reservoir (Negrete-Aranda et al., 2021). Our results extend this connectivity to the microbial communities between fields.

To complement this community wide characterization, we additionally conducted a targeted analysis of the ASV-level diversity of members within the Thermotogota, a deeply branching bacterial phylum known to have cultured representatives spanning a broad temperature range. Analysis of the new vent field, JaichMaa ‘ja ‘ag, and previously described Auka site revealed distinct patterns in the composition, distribution, and abundance of Thermotogota ASVs across the basin, highlighting the potential for distinct ecological adaptations between lineages in this physico-chemically heterogeneous system.

Diversity of Thermotogota in the S. Pescadero Basin hydrothermal vents

Thermotogota are a deep-branching bacterial phylum that includes hyperthermophilic, thermophilic and mesophilic heterotrophs adapted to organic-rich, anoxic environments.

Characterized members of Thermotogota all have a distinctive sheath-like toga structure surrounding the cell (Huber et al., 1986). Cultured representatives are all anaerobes and capable of fermentation or growing organotrophically with thiosulfate or elemental sulfur (Conners et al., 2006; Itoh et al., 2016). Members of this phylum have been detected from hydrothermal vents, geothermal environments, and petroleum reservoirs worldwide (Bhandari & Gupta, 2014b; Gupta & Bhandari, 2011; Itoh et al., 2016; Orphan et al., 2000; Reysenbach et al., 2013; Zeng, Alain & Shao, 2021). Thermotoga maritima MSB8 was the first described member of the Thermotogota isolated from coastal geothermal sediments in Italy, with an optimum growth temperature of 80 °C (Huber et al., 1986). Subsequent characterization of other Thermotogota has expanded the physiological range to include mesophilic and thermo-acidophilic isolates (Belahbib et al., 2018).

All five families (Thermotogaceae, Fervidobacteriaceae, Kosmotogaceae, Petrotogaceae, and Mesoaciditogaceae; Bhandari & Gupta, 2014a) have been reported within hydrothermal vents, highlighting the diversity of this phylum in these extreme environments, and matching our observations of their distribution across the Auka and JaichMaa ‘ja ‘ag sites. Only a single genus of the family Fervidobacteriaceae (Thermosipho) has been described in marine hydrothermal environments. This genus appears to have a widespread hydrothermal distribution including Djibouti, Africa (Huber et al., 1989), deep-sea hydrothermal vents in the Lau Basin (Antoine et al., 1997), the Iheya Basin (Takai & Horikoshi, 2000), the Mid-Atlantic Ridge (Podosokorskaya et al., 2011; Urios et al., 2004), the Suiyo Seamount in western Pacific Ocean (Kuwabara et al., 2011), and in Guaymas Basin (Podosokorskaya et al., 2014). While we chose to focus on the most abundant Thermotogota ASVs as an example of the spatial distribution patterns of this phylum, phylogenetic analysis of all ASVs suggests that even greater diversity of the Thermotogota occurs within the S. Pescadero vents (Fig. S4).

ASVs belonging to the families Thermotogaceae, Fervidobacteriaceae, Kosmotogaceae, and Mesoaciditogaceae were the most abundant members of the Thermotogota phylum in the hydrothermal vent fields of Auka and JaichMaa ‘ja ‘ag. Surprisingly, members of the Petrotogaceae, previously described from deep, hot oil reservoirs and vents (Alain et al., 2002; L’Haridon et al., 2019; Orphan et al., 2000; Postec et al., 2005), were only detected at low abundance despite the presence of sediment-associated hydrocarbons in the basin (Speth et al., 2022).It is possible that members of the more abundant Kosmotogaceae, which also occur in deep subsurface oil reservoirs (DiPippo et al., 2009; Pollo et al., 2017; Swithers et al., 2011) may occupy hydrocarbon-rich niches in the S. Pescadero Basin rather than Petrotogaceae.

Thermophily appears to be a phylogenetically conserved trait among most Thermotogota families. For example, the Thermotogaceae consist of thermophilic and hyperthermophilic bacteria with optimum growth temperatures (OGTs) ranging from 60 °C to 80 °C (Zhaxybayeva et al., 2019) with some members (e.g., Thermotoga maritima and Thermotoga neapolitana) as high as 90 °C (Huber et al., 1989). In contrast, Mesoaciditogaceae are moderately thermophilic and acidophilic (pH 6.0), growing optimally around 55 °C to 60 °C with a GC content of 39.9–40.6 mol% (Itoh et al., 2016). The distribution of Thermotogota families within the JaichMaa ‘ja ‘ag vent field in some samples appears to support these predictions. For example, the relative abundance of the ASV Thermotogaceae_1 was high at the Abuelita site, with the previously reported OGT values for this family falling within the recorded sediment temperatures for the site (Data S5). Differences in thermotolerance among the Thermotogota may explain some of down-core shifts inf family-level representation, where sediment cores PC6 and PC1 from Abuelita, showing moderate gradients in Mg, were dominated by Mesoaciditogaceae_1 near the seafloor by up up to 94% and transitioned to Thermotogaceae_1 or Thermotogae_2, respectively with depth. However, inferred temperature tolerance among Thermotogota does not appear to explain the complete distribution patterns of this phylum; for example, the Thermotogota in some cores (e.g., PC6 and PC7 from south of Z) were dominated by a single Mesoaciditogaceae ASV from top to bottom, although the total abundance of Thermotogota changed from 0.5% to 30% along with large gradients in Mg concentration. These observations likely point to factors other than temperature controlling community composition. Or perhaps this Mesoaciditogaceae ASV distribution may reflect greater diversity in optimal growth temperatures than currently recognized for this family, similar to other Thermotogota groups (e.g., Kosmotogaceae with an OGT range from 37 °C and 70 °C, Nesbø et al., 2021). The overall agreement between family-level OGT values and observed distribution patterns point to temperature and vent fluid associated physico-chemical conditions as a major determinant of the Thermotogota distribution in Auka and JaichMaa ‘ja ‘ag vent fields. We also note that as this is a DNA-focused study, some of the recovered 16S rRNA gene diversity may also be affiliated with non-viable cells (Li et al., 2017). As such, we cannot rule out the possibility that the documented changes in Thermotogota ASVs down core may be attributed to mixing from bioturbation or active hydrothermal circulation in the sediments. Future analyses incorporating RNA analysis or targeted fluorescence in situ hybridization microscopy will assist with advancing our understanding of potential temperature-based niche differentiation among Thermotogota families in sedimented hydrothermal vents.

In addition to the physiological characterization of cultivated isolates, genome sequencing and environmental metagenomic assembled genomes (MAGs) can be used to infer optimal growth temperature based on GC content and amino acid composition for microbial groups that have not yet been physiologically characterized (Li et al., 2019; Sauer & Wang, 2019). Previously, Speth et al. (2022) used MAGs reconstructed from sediment-hosted microorganisms at the Auka vents for OGT predictions of representative bacterial and archaeal lineages. Eleven of these MAG’s were affiliated with the Thermotogota, with estimated completeness ranging from 60 to 100 and estimated contamination ranging from 1.02% to 5.93%. OGT predictions were performed on these Auka MAGs and correlated with phylogeny, containing representatives from 3 of the 4 Thermotogota families along with other as yet uncharacterized lineages (Speth et al., 2022). This dataset was used to provide additional ecological context to the distribution of ASVs across the 246 samples in this study. Kosmotogaceae MAGs from Auka had a predicted OGT value of 44 °C, consistent with characterized cultured isolates (Nesbø et al., 2021) and the lateral and down core distribution of Kosmotogaceae ASVs distribution in this study (Fig. 5A). The predicted OGT values of Thermotogaceae MAGs ranged between 46 °C to 65 °C, while Fervidobacteriaceae had a predicted OGT value of 53 °C. Both families had genome-based predicted OGT values lower than reported for cultured representatives (Bhandari & Gupta, 2014a; Farrell et al., 2021; Zhaxybayeva et al., 2019), possibly pointing to the potential for mesophilic members within the Thermotogales. This is consistent with previous reports suggesting that distinct Thermotogales lineages are distributed in low-temperature environments but are likely underreported (Nesbø et al., 2010). The remaining five MAGs within the Thermotogota phylum belonged to family level lineages without cultured representatives and thus lack empirical data to compare with the genomic OGT predictions. OGTs for these uncultured lineages ranged between 42 °C and 51 °C, supporting the potential for a greater diversity of mesophilic representatives within the Thermotogota across the S. Pescadero Basin.

Biogeography and distribution of Thermotogota in the S. Pescadero Basin

We found 18 ASVs related to Thermotogota with different patterns of distribution and abundance across the S. Pescadero Basin, suggesting the ability of this phylum to inhabit the diverse niches in subseafloor hydrothermal vent environments. These diverse niche environments provide opportunities for the colonization and adaptation of microbial lineages (Anderson, Sogin & Baross, 2015). Our down-core trends on the centimeter scale extends previous insights into the occurrence patterns of distinct microbial lineages within hydrothermal vent sediments (Lagostina et al., 2021; Zhang et al., 2021).

Different ASVs are dominant at each Pescadero Basin site, yet several of the same ASVs were detected across several samples. This overlap suggests that the hydrothermal vent fields draw from a shared pool of ASVs, i.e., dispersal bridges the two vent fields. Indeed, several recent studies of the vent systems in the Gulf of California (S. Pescadero Basin and Guaymas Basin) have come to similar conclusions based on the microbial composition (Laso-Pérez et al., 2023; Speth et al., 2022; Wu et al., 2022) and geophysical attributes (Negrete-Aranda et al., 2021).

Assuming both Auka and JaichMaa ‘ja ‘ag have sufficiently low barriers to dispersal, they represent an ecologically interesting pair as hydrothermal fluids appear more diffuse in JaichMaa ‘ja ‘ag than Auka (inferred from magnesium concentrations). Indeed, Auka had the highest temperatures of 128 °C at 30 cmbsf compared to a maximum temperature of 86 °C in JaichMaa ‘ja ‘ag. Thus, these environmental differences could play a role in shaping microbial communities at small scales (e.g., adjacent cores), contributing significantly to the spatial variation observed in microbial diversity as summarized by Martiny et al. (2006).

Beyond thermal variation, other drivers of community composition have been documented in large-scale geological settings (Galambos et al., 2019) where differential enrichments of sulfide, hydrogen, methane, or other hydrocarbons can occur (Amend et al., 2011; Brazelton et al., 2006; Teske, 2020). Biotic drivers are also well-documented drivers of community structure in marine sediments (Holden & Adams, 2003; Thomas et al., 2021; Urich et al., 2014), and indeed recent work in the S. Pescadero Basin has highlighted the prevalence and eco-evolutionary impact of phage communities, associated with ANME-1 archaea or Heimdallarchaea (Laso-Pérez et al., 2023; Wu et al., 2022).

Conclusions

In this study, we report the microbial ecology of the recently discovered hydrothermal vent field of JaichMaa ‘ja ‘ag located in the Southern Pescadero Basin. The proximity of vents within the Pescadero Basin offers the opportunity to investigate complex microbial community dynamics and ecosystem functioning at different scales. Our ecological analyses comparing the diversity of ASV composition with the physicochemical characteristics of the basin suggest that both Auka and JaichMaa ‘ja ‘ag are components of a shared hydrothermal system. The microbial distribution patterns of Thermotogota, a phylum known for thermophily, in the Pescadero Basin appear to be influenced in part by the temperature gradients in the hydrothermal fluids. The identification of differences in the prevalence of the Thermotogota assemblages highlights the intricate variables within the hydrothermal ecosystem, underscoring the need for additional research to fully understand them. JaichMaa ‘ja ‘ag represents a remarkable advancement in exploring hydrothermal vent ecosystems at the Gulf of California, with the potential to enhance our knowledge of the physical, geochemical, and biological processes that shape the microbial communities in the deep-sea biosphere.

Supplemental Information

Supplemental Information 1 Geochemical and 16SrRNA Analyses.

This section encompasses the results of comprehensive geochemical analyses and 16SrRNA sequencing conducted on sediment cores collected from the Pescadero Basin. The geochemical analyses provide insights into the elemental composition and characteristics of the sediments, while the 16SrRNA sequencing data reveals the microbial diversity within the sampled cores.

Supplemental Information 2 Processing Code and Statistical Analyses.

Specific code employed for processing the genetic sequences obtained from the samples. This includes any algorithms or computational methods used in the analysis.

Supplemental Information 3 Sequence Counts.

Raw counts derived from the sequencing process. These counts quantify the occurrences of specific genetic sequences, offering a quantitative measure of microbial abundance and diversity in the sediment samples.

Supplemental Information 4 Taxonomy Assignment for ASVs.

Taxonomic classification assigned to the Amplicon Sequence Variants (ASVs). These taxonomic assignments elucidate the identity and categorization of the microbial communities present in the sediment cores.

Supplemental Information 5 Sample list and physicochemical data obtained from Pescadero Basin during expedition FK181031.

Temperature data were measured in situ. Ammonium and magnesium were determined by ion chromatography. The detailed information about the environmental conditions under which the sediment samples were collected, offering context to the geochemical and microbial analyses.

Supplemental Information 6 Microbial diversity in the S. Pescadero Basin hydrothermal vent fields.

The relative frequencies of phyla derived from the 16S rRNA amplicon sequence variants found in the hydrothermal vent fields of Auka and JaichMaa ’ja ’ag.

Supplemental Information 7 Microbial diversity correlated with hydrothermal fluids beneath the sediments in the Pescadero Basin hydrothermal vent fields.

Microbial diversity correlated with hydrothermal fluids beneath the sediments in the Pescadero Basin hydrothermal vent fields. The most abundant Phyla with respect to magnesium [mM] in (A) Auka, (B) JaichMaa ‘ja ‘ag, and (C) Juwak Yuum with color coding based on ammonium and sulfide [mM]. All taxa correspond to Bacteria except for Crenarchaeota which corresponds to Archaea. The unclassified group could include both domains.

Supplemental Information 8 Microbial structure and composition of the phylum Thermotogota in the Auka and JaichMaa ‘ja ‘ag hydrothermal vent fields.

The relative frequency is no normalized to total Thermotogota abundance.

Supplemental Information 9 Phylogenetic comparison of Thermotogota ASVs from S. Pescadero Basin with reference 16S rRNA sequences from the Genome Taxonomy Database.

the tree illustrates the evolutionary relationships between Thermotogota ASV sequences obtained from the study (highlighted in red) and reference sequences.

We thank Dr. Stephanie Connon (Caltech) for assistance with preparation of samples for iTAG sequencing and express our gratitude to the Schmidt Ocean Institute’s R/V Falkor and ROV SuBastian pilots and crew and the shipboard science party of FK181031 for their invaluable contributions during the oceanographic expedition. We extend our appreciation to the California Institute of Technology and members of the Orphan lab for providing guidance and the necessary resources for M.P-S. to conduct this research. M.P-S. also acknowledges the Agouron Simons International Geobiology Course for the opportunity to learn bioinformatics skills. Additionally, we acknowledge the Astronomy Institute of UNAM for providing essential facilities as an external doctoral student, and to the Astrobiology Laboratory for their unwavering support throughout the project’s development. We additionally express our gratitude and appreciation of the mentorship, support, and scientific inspiration of the late Dr. Jan Amend, hydrothermal vent geobiologist and director of the center for Dark Energy Biosphere Investigations (C-DEBI). We express our sincere gratitude to William Brazelton and the anonymous reviewers for their valuable feedback, which greatly contributed to enhancing the quality of this article.

Additional Information and Declarations

Competing Interests

Author Contributions

Field Study Permissions

DNA Deposition

Data Availability

The authors declare that they have no competing interests.

Manet E. Peña-Salinas conceived and designed the experiments, performed the experiments, analyzed the data, prepared figures and/or tables, authored or reviewed drafts of the article, and approved the final draft.

Daan R. Speth conceived and designed the experiments, performed the experiments, analyzed the data, prepared figures and/or tables, authored or reviewed drafts of the article, and approved the final draft.

Daniel R. Utter conceived and designed the experiments, performed the experiments, analyzed the data, prepared figures and/or tables, authored or reviewed drafts of the article, and approved the final draft.

Ronald M. Spelz conceived and designed the experiments, performed the experiments, analyzed the data, prepared figures and/or tables, authored or reviewed drafts of the article, and approved the final draft.

Sujung Lim conceived and designed the experiments, performed the experiments, authored or reviewed drafts of the article, and approved the final draft.

Robert Zierenberg conceived and designed the experiments, authored or reviewed drafts of the article, and approved the final draft.

David W. Caress conceived and designed the experiments, authored or reviewed drafts of the article, and approved the final draft.

Patricia G. Núñez conceived and designed the experiments, authored or reviewed drafts of the article, and approved the final draft.

Roberto Vázquez conceived and designed the experiments, authored or reviewed drafts of the article, and approved the final draft.

Victoria J. Orphan conceived and designed the experiments, analyzed the data, prepared figures and/or tables, authored or reviewed drafts of the article, and approved the final draft.

The following information was supplied relating to field study approvals (i.e., approving body and any reference numbers):

Sample collection permits were granted by la Dirección General de Ordenamiento Pesquero y Acuícola, Comisión Nacional de Acuacultura y Pesca (CONAPESCA: Permiso de Pesca de Fomento No. PPFE/DGOPA-200/18) and la Dirección General de Geografía y Medio Ambiente, Instituto Nacional de Estadística y Geografía (INEGI: Autorización EG0122018), with the associated Diplomatic Note number 18-2083 (CTC/07345/18) from la Secretaría de Relaciones Exteriores - Agencia Mexicana de Cooperación Internacional para el Desarrollo / Dirección General de Cooperación Técnica y Científica.

The following information was supplied regarding the deposition of DNA sequences:

The Auka 16S rRNA gene amplicon sequencing data were previously described in Speth et al. (2022) and is available in NCBI: PRJNA713414.

The newly reported samples are available at NCBI: PRJNA1045058; SAMN38428873 through SAMN38428956.

The following information was supplied regarding data availability:

The raw data is available in the Supplemental Files.

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
