# Peer review of "Thermotogota diversity and distribution patterns revealed in Auka and JaichMaa ‘ja ‘ag hydrothermal vent fields in the Pescadero Basin, Gulf of California"

_PeerJ, doi:10.7717/peerj.17724_

## Round 0.1 · original submission · Major Revisions

Please provide a point-by-point rebuttal letter to all of the reviewer comments along with your revised manuscript.

·

Basic reporting

Peña-Salinas et al. report a microbial diversity survey of a newly discovered hydrothermal vent field in the Gulf of California. Previous studies have been conducted at a nearby vent field (Auka, only 2 km away), and this study is the first report of microbial diversity from the new field, named JaichMaa 'ja 'ag.

Experimental design

As an initial report of microbial diversity, it is necessarily descriptive, but the authors do a nice job of presenting the data in visually appealing and informative figures. The study seems to have been initially motivated by questions regarding the distribution of thermophiles within the vent field, but the results contained little to no clear correlations with temperature. Instead, the authors report some intriguing connections with magnesium, ammonia, and other variables. I think this makes sense, as temperature is very difficult to measure in situ, especially on a microbial scale, as the authors point out. All of this is perfectly fine, but I do think that parts of the paper, especially the abstract, should be rewritten to reflect these results, focusing on the geochemical trends, rather than temperature.

Validity of the findings

In general, this is a well-executed, clearly described study about an exciting new hydrothermal vent field. I have only a few minor comments below and otherwise would certainly support publication of a moderately revised version.

Additional comments

Specific comments:
Title: The title mentions "diffuse fluids", but all of the analyses were conducted with sediment samples. This is very confusing. I suggest swapping "diffuse fluids" with "sediments" in the title, or simply omit that part of the title, so that it reads "... patterns revealed in Auka and JaichMaa 'ja 'ag hydrothermal vent fields.."

Abstract: I'm not a big fan of the first sentence because I don't think thermophily on its own is an "extreme" trait, and certainly the growth temperatures mentioned in this study do not approach any of "life's boundaries on Earth". I think there are many reasons why these organisms and this environmental system are super interesting, and the possibility that they might like to live in conditions slightly warmer than body temperature is not one of them. I may be wrong about this - I was actually a bit confused about the inferred growth temperatures in this study because none of the figures report temperature. I think the absence of temperature in the figures and the key conclusions also argues for a first sentence that focuses on something else.

Figure 1: The font of the labels in the inset of Figure 1 are very difficult to read, even after zooming in. The label fonts in the main figure (A and B) could be much larger too.
Figure 2 is very nice. I appreciate the high-quality map providing context for the cool photos.
Figures 3-5 are good visual representations of results that are often very difficult to present.
Supplemental Data: The additional tables with original data are very much appreciated. I'm happy to see this becoming more and more common in microbial diversity papers.

Reviewer 2 ·

Basic reporting

Peña-Salinas et al describe the distribution and diversity of Thermatoga in the newly discovered hydrothermal setting of JaichMaa ‘ja ‘ag (Gulf of California), using 16S rRNA amplicon sequencing. The authors collected push cores and extracted DNA from shallow sediments along the north-south axis of the vent, from sites that have different degree of hydrothermalism. Further, the authors performed a comparative analysis using community 16S rRNA Thermatoga data from hydrothermal systems that are in vicinity with JaichMaa ‘ja ‘ag, (e.g. Auka vent).
Overall, the study is well written; it provides sufficient field background, and the research question is well defined. I would like to endorse it for publication.

As general comment, DNA (16S rRNA, metagenomes, SIP etc) and RNA analyses from hydrothermal systems are always precious as they extend our understanding on life in early Earth, can help us describe and extend biodiversity and also understand the mechanisms of microbial survival under such challenging conditions.

Some suggestions for the authors are to comment that Thermatoga were a small fraction from the identified prokaryotic community in JaichMaa ‘ja ‘ag (if I am not mistaken 152 Thermatoga ASVs out of 26,162 total ASVs), and that the rest of the identified JaichMaa ‘ja ‘ag ASVs (SD4 data Table) belong to commonly identified taxa on hydrothermal settings (e.g., Desulfo, ANME-1 etc). Also, another suggestion for the authors is to include in the potential explanations in their discussion (lines 465-478 down-core changes of Thermatoga) that 16S rRNA captures cells which are not necessarily alive, hence, it is possible that some of the down-core Thermatoga changes to be the result of cells transported with active hydrothermal circulation. Likewise, to comment that alive Thermatoga may have the genomic potential/flexibility to respond to geochemical and thermal gradients and this could explain their presence at conditions which were not anticipated to be present.

Finally, it would be beneficial for the reader if authors included in the material and methods which elements were analyzed with Ion Chromatography (either cite the SD metadata 1 Table or make a comment that from the IC data generated we show XX, YY which are relevant to our field study); to mention that the sediments analyzed are collected from single core per site (no biological replication, unless if I missed it). Single core is fine and no argument here because hydrothermal sediments are difficult to collect and also are challenging for DNA and RNA extractions.

Some minor comments:
line48: remove “Add your introduction here”. Same with other places.

Experimental design

See above

Validity of the findings

See above

Reviewer 3 ·

Basic reporting

This study analyses occurrence patterns of Thermotoga-affiliated ASVs at newly discovered vent sites in the southern Pescadero Basin, and attempts to infer biogeographical patterns from the observed distributions and habitat preferences of these ASVs in hydrothermal sediment cores. In brief, hydrothermal sediments marked by low Mg content contain high relative proportions Thermotoga ASVs, and these ASVs show local compositional variation that may be linked to chemical and temperature factors but these are hard to pin down specifically. “Biogeographical” patterns of Thermotoga ASVs remain hard to explain, not only because of uneven metadata but also as a consequence of conceptual confusion. High degrees of habitat connectivity among adjacent vent sites rule out biogeography in the sense of spatial separation and geographical isolation of genetically distinct populations. Instead, local chemical and temperature conditions at each sampling site select from a pool of widely shared microbial populations, and this seems to be the case here. However, before attempting to infer habitat preferences (a secondary task), it would be interesting to learn more about Thermotoga ASV diversity in the Southern Pescadero vent sites, and to show some phylogenies that introduce and define the diverse Thermotoga lineages that populate these hydrothermal sediments. Finally, the newly discovered JaichMa’ja’ag vent sites look very interesting in themselves, but are presented poorly (tiny image panels in Figure 2). Of course, reviewers are not supposed to rewrite papers, but this reviewer feels strongly that a descriptive study of the new vent sites, their geochemistry and their microbial populations would be a beautiful and strong contribution (in this regard, the supplements contain substantive data), whereas the Thermotoga distribution patterns are of secondary interest (they exist, but are hard to explain) and could be documented in supplements. Since this manuscript is a first submission, this revision could be an opportunity for adjusting the relative weight of these manuscript components.

Details:
Line 42: ... that point to...
Lines 48 to 62 are very generic background on uncultured microbial life, the solar system, hydrothermal vents etc. that do not add significantly to this paper. these lines could just as well be replaced by a short, concise introduction on microbial diversity and biogeography at hydrothermal vents.
Lines 65 to 86 read like a citation-heavy introduction to the geology of the Gulf of California. Please keep in mind that this is a microbiology manuscript, and the introduction should therefore limit the geology to the minimum that is required to set the stage for the Pescadero Basin hydrothermal vents. Brevity is a virtue...
Throughout the introduction and the manuscript in general: Basin or basin?
Line 129: “data” is a plural noun, “datum” is the singular; adjust verbs accordingly
Line 137ff: Please include a sample table that lists the hydrothermal sediment cores sampled, coring depths, lat/long coordinates, in-situ temperatures, sampling intervals used for DNA extraction and sequencing, and so on. I see that the supplements contain a large metadata table, but readers will be grateful for a short version (within the manuscript!) that provides essential context for this particular study.

Line 158- 196. The geochemical analyses could be summarized and referenced, and this elaborate text-heavy version could be placed into supplementary text.

Line 266 ff: The description of the recently discovered JaichMa’ja’ag vent field and its sampling locations looked more compelling than the following attempt to obtain “biogeographical” information from the distribution pattern of Thermotoga ASVs.

Line 348 ff: To increase the scientific mileage of this ASV dataset, you could generate a comprehensive phylogeny of these Thermotoga ASVs (beyond what is shown in Fig 5A). The 400 bp fragments are not ideal, but it would be doable. Or you could generate a phylogenetic framework using longer literature sequences, and then place the shorter ASVs into this frame.

Line 393-395: I really do not understand this statement.

Line 501 ff: In this discussion, I have the general impression that the distribution patterns of Thermotoga ASVs are interpreted with a heavy hand, and too much is being read into the observed distributions. After all, these are just amplicon frequencies, subject to numerous potential biases during DNA extraction and ASV amplification. Not every single feature of ASV distribution needs to reflect some external factor. Interpreting the ASV patterns is particularly ambiguous since the sampling site metadata appear have irregular gaps, in particular temperature. Hydrothermal circulation patterns can also mess up ASV distributions.

The discussion on “biogeography” should be more precise. Over short distances of a few hundred meters (or 2 km), microbial cells are easily distributed between different hydrothermal spots. Habitat connectivity among adjacent vent sites is very high, and biogeography in the sense of spatial separation and geographical isolation of genetically distinct populations does not exist (for a relevant case study, see Meyer et al. 2013 Frontiers in Microbiology 4:207; doi: 10.3389/fmic.2013.00207). Instead, local chemical and temperature conditions at each sampling site select from a pool of shared microbial populations. For conceptual clarification, see Martiny et al. 2006. Microbial biogeography: putting microorganisms on the map. Nat Rev Microbiol 4, 102–112 doi: 10.1038/nrmicro1341.

Line 532: Are you sure that these electron acceptors (oxygen, nitrate, oxidized metals) are relevant in highly reducing sediments? Guaymas Basin sediments are completely anoxic and nitrate-depleted below the upper few millimeters, and these hydrothermal sediments in the southern Pescadero Basin are most likely quite similar.
Line 534: Infection with bacteriophages is not a predator-prey relationship; predators use their prey as food, and they do not reprogram their prey to generate more predators...
Line 554: thermophily

Figure 2: Is it possible to re-design this figure so that the fascinating in-situ photos are enlarged? At present they are the size of postage stamps, but I assume the image quality is sufficient to allow reproduction in a larger format.

Are the magnesium concentrations used for plotting Figures 3 and 4 always porewater concentrations, or did you include datapoints from directly sampled vent fluids?

Figure 4 does not show the relative abundance, it shows the relative frequency of Thermotoga ASVs among all ASV sequences.

Figure 5: Please move panel B below panel A; in this way you can enlarge both panels and make them more accessible. The sample (cmbsf) line at the bottom of panel A is too small to be readable.

The references need attention; the formatting is quite haphazard and some references are incomplete:
Line 698: journal information is missing.
Line 815; Neumann et al. is now published: Neumann et al. 2023, Basin Research 35:1308-1328. DOI: 10.1111/bre.12755

Experimental design

I would suggest to make good use of the broad scope of PeerJ in environmental and biological sciences, and re-tool this manuscript into a primary description of the new vent sites, their geochemistry and microbiology, with some comparison and contrast to the Auka vents. Use the same data but play to your strengths! The Thermotoga "biogeography" is a problematic issue; obviously there is some site-specific environmental selection going on, but the mechanisms seem to be unclear, and ASV distributions alone have limited explanatory power. By reducing the relative weight of this topic, it is also possible to throw out a lot of speculation that leads nowhere (viruses, "predation", unknown electron acceptors, etc).

Validity of the findings

As far as I see, the metadata for the new vent sites are sound, and I would not hesitate to put a handy selection of them into the manuscript, as tables or plots. They are more valuable than you think, and there is no reason to hide them and to use them only as background for the Thermotoga story (which is not so compelling after all).

Additional comments

To summarize - think about the most effective story that you can tell, something that people really want to read. New vent sites in the Pescadero Basin are wonderful; you have excellent maps and images, good geochemistry, some temperature data, and acceptable ASV data that introduce the microbial diversity of these sites. There is no need to focus on the problematic "biogeography" aspect of the entire story where a good answer or solid conclusion is impossible; this seems like needless self-flagellation.

---

## Round 0.2 · Minor Revisions

A minor revision is required, so please consider in full details these last suggestions.

Reviewer 3 ·

Basic reporting

In general I am OK with the revision. The beautiful photo panels in Figure 2 are enlarged, a phylogeny of Thermotogota ASVs is provided, and the overstretched biogeographical interpretations are toned down somewhat.
However, the conclusion paragraph is still marred by overextended language on “complex understanding” and evolutionary insights that should be replaced by more accurate assessments (see box 3, validity of findings). In contrast to the somewhat inconclusive biogeographic inferences on Thermotogota, the findings about the new JaichMaa ja ag vent field are indisputably exciting, and could easily be highlighted in the conclusion.

Text and Figure details require attention:

Lines 316, 323: Here you should refer to Figure 3a.
Line 344 and 345: Figures 4A and B have nothing to do with correlation coefficients; they simply show the relative contribution of Thermotogota ASVs at Auka and JaichMaa ja ag, in different samples plotted against magnesium concentration on the x-axis.
Line 348 to 349: very convoluted, I really needed to look at Figure 4 to understand this at all. Here you can simply say that Magnesium concentrations at Auka started at low concentrations near 1 mM, whereas at JaichMaa ja ag they started at 20 mM.
Line 455: you mean “sheath-like”
Line 460: Does “this group” refer to the genus Thermosipho, or to the entire family Fervidobacteraceae?
Line 460: better: “…a widespread hydrothermal distribution…”
Line 460: What is “Obock”? the in abstract to Huber et al. 1989, it says “a marine hydrothermal area near Obock, Djibouti…)
Lines 474 and 475 should not be separated by a paragraph break; instead the new paragraph should start at line 478 (Thermophily appears…”)
Line 512 and 522: MAGs (plural does not require an apostrophe)
Line 520: This dataset was used…
Line 563: “of phage” should be specified; this is a complex phage community, often associated with ANME-1 archaea

Figure 2: Is it possible to include lat/long designations along the edge of this bathymetry? This will be the first published detailed bathymetry of this vent region, and lat/long designations are essential for a reference map.

Figure 3 legend: ”… and the third vent site, Juwak Yuum…”
Try to replace acronyms such as FK181031 (presumably a cruise number?) with something more accessible.

Experimental design

no comments

Validity of the findings

The concluding assessment of these findings needs some work

At present, line 574 to 576 seems to be boilerplate language that hides the simple fact that no particular environmental factor explains the distribution of Thermotogota ASVs. This study does not convey a “complex understanding” but a complex situation where numerous environmental variables make clear inferences really difficult.
Line 577-578, this statement about the evolution of thermophily in the Thermotogota lineage, is exaggerated and not justified by the contents of this paper. The statement would be appropriate for a phylogenomic study of thermophily within the Thermotogota, but what we have here are inferences based on the environmental distribution of short 16S gene fragments.
As a more realistic summary, uncharacterized diversity within the Thermotogota and divergent trends in their environmental distribution make it difficult to link environmental variables and lineages; more genomic and cultivation work is necessary to provide a solid basis for these connections in the future.
However, the discovery and documentation the new JaichMaa ja ag vent field provides a good opportunity to say something positive in the conclusion, and to end this paper on a strong note.

Additional comments

no further comments

---

## Round 0.3 · accepted · Accept

Thank you for your patience during the revision process and for following the reviewers' comments.